# DROPOUT AND THE OUTLIERS: COULD TRANSFORMERS OVERCOME THEIR SINGLE POINTS OF FAILURE?

**Nour Hezbri**[*], **Gilles Bareilles & El-Mahdi El-Mhamdi**
CMAP, École Polytechnique, France
`{name}.{surname}@polytechnique.edu`

## ABSTRACT

Due to their complex structure, Transformer architectures give rise to curious empirical phenomena. One such phenomenon recently attracted significant attention: the formation of disproportionately large attention, weight and activation values during training, often called *outliers*. Hardware, security and other favorable properties make these extreme values undesirable for robust architectures. While many recent works observed the issue of outliers in transformer-based models, rarely did they consider Dropout's effect, an algorithm that was initially designed to increase neural networks' robustness. In this work, we provide a systematic assessment of the effect of Dropout on the formation of outliers across different modality, architecture and optimization choices. We show in our setups that Dropout helps reduce the outliers on average but does not suppress them completely. Our findings provide a paradoxical view, that contrasts with the folklore belief that Dropout tends to equalize values across the network. It also raises important questions on the implicit bias of Dropout in transformer-based models within certain optimization and architectural choices.

## 1 INTRODUCTION

For the last few years, Transformers (Vaswani et al., 2017) became a dominant architecture in Deep-Learning, enabling models to achieve state-of-the-art results in a variety of domains. From residual connections to layer normalization and attention layers, the complex transformer architecture builds on most of the important architecture and optimization related advances of the past two decades in Machine-Learning, on top of foundational recipes such as the Multi-Layer Perceptron (MLP), favoring the emergence of various interesting empirical phenomena.

One such phenomenon is the formation of outliers, across the transformer-based model's attention maps (Gu et al., 2025; An et al., 2025; Darcet et al., 2024), weights (Yu et al., 2024a; An et al., 2025) and activations (Kaul et al., 2025; An et al., 2025). Outliers generally refer to rare, larger than average values, that appear early in pre-training and persist throughout training, inference and finetuning. They appear in bidirectional (Kovaleva et al., 2021), autoregressive Language models (An et al., 2025), as well as Vision models (Darcet et al., 2024) and Multimodal models (Kang et al., 2025).

The outlier emergence phenomenon particularly challenges the robustness of transformer-based models, despite their overparametrization and redundancy (Men et al., 2024; Gromov et al., 2025). If we view a Neural Network as a distributed system (El Mhamdi & Guerraoui, 2017; Wu et al., 2024), these outliers represent single points of failure (Lynch, 1996), whose failure would disrupt the model and cause potentially significant performance degradation (Sun et al., 2024). Hardware issues and cybersecurity concerns, such as bit-flip attacks (Rakin et al., 2019), make outliers even more interesting as an attack surface of modern Machine-Learning algorithms.

In its modern adoption (Srivastava et al., 2014), the *Dropout algorithm* is often thought of as a regularization means to mitigate overfitting, and improve the generalization ability of the trained model. Dropout consists in randomly deactivating a subset of neurons with a probability rate $p$. The same recipe was proposed earlier by Kerlirzin & Vallet (1993) and Kerlirzin & Refregier (1995),

---

[*]Correspondance to nour.hezbri@polytechnique.edu

under the name "mortality algorithm", to enhance the network's resilience to single[1] points of failure, by simulating the loss of certain components of the network during training.

Hence, Dropout appears as a natural candidate to mitigate the presence of outliers. Two more recent observations add credence to this hypothesis. First, Mianjy et al. (2018) shows that Dropout's implicit bias has a rebalancing effect on the weights for a single hidden layer linear NN: roughly, the mass of the weights is distributed more uniformly. One could then reasonably assume that a similar effect could extend to more general multi-layer, highly non-linear models. Second, He et al. (2024) relates the Outlier Features to bad signal propagation during training and rank collapse phenomena that transformer-based models often suffer from (Anagnostidis et al., 2022; Saada et al., 2025). Dropout helps reduce the correlations between input features, improving signal propagation (Kedia et al., 2024), hence its relevance as a potential mitigation strategy.

However, although key modern architectures such as BERT (Devlin et al., 2019) and Vision Transformers (ViTs) (Dosovitskiy et al., 2021) were originally trained with a $p = 0.1$ Dropout rate, Dropout's popularity and usage is decreasing in modern training pipelines (Jiang et al., 2023; Grattafiori et al., 2024). This is arguably due to the ever-increasing size of training-data (Schaul et al., 2023), which decreased overfitting and thus the need for regularization.

Hence, the impact of Dropout on the presence of outliers was not thoroughly studied so far, despite its intuitive relevance in outliers suppression. Indeed, we are aware of only one work on outliers (He et al., 2024) that explicitly discusses Dropout's effect alongside proposed architectural modifications, without the systematic study of the effect of Dropout that our work provides. Surprisingly, we observe across various setups that Dropout does indeed, on average, slightly reduce the outliers. However, its impact is marginal or even negligible for some particular architecture/optimization choices. Our findings provide a paradoxical view, that contrast with the folklore belief that Dropout tends to equalize values across the network. It also raises important questions on the implicit bias of Dropout in transformer-based models within certain optimization and architectural choices.

## 2 NOTATION AND DEFINITIONS

### 2.1 TRANSFORMERS

We use the following notations: $\tau$ denotes a threshold value, $H$ denotes an activation *i.e.,* a transformer layer output taken after the residual connection, $A$ denotes the attention map, resulting from the application of the softmax operator on the key-query dot product, and $W$ denotes a weight tensor. We also use $T$ to denote the sequence length, $d$ for the embedding dimension, $H$ for the number of transformer heads per block and $B$ the batch size. More notational details can be found in Section B of the Appendix.

### 2.2 OUTLIERS

In the following, we adopt the definitions of An et al. (2025) for the categories of outliers. Later on, we fix $\tau = 5$ in these definitions, for our experiments.

- **Activation Outliers:** At layer $\ell$, for transformer layer outputs $H^\ell \in \mathbb{R}^{B \times T \times d}$, the set of $\tau$-activation outliers $O^\tau_{\text{activation}}$ is defined as:

$$O^\tau_{\text{activation}} = \{(i, j, k) \mid |H_{i,j,k}| > \tau \cdot \mu_h\},$$

where

$$\mu_h = \frac{1}{B \cdot T \cdot d} \sum_{i,j,k} |H_{i,j,k}|$$

is the mean absolute value of $H^\ell$.
This definition coincides with the notion of *"super activations"* exhibited in Yu et al. (2024a) and that of *"massive activations"* of Sun et al. (2024).

---

[1]"Single" points of failure are not necessarily individual components of a system, these could also be pairs or small subgroups of components, whose joint failure disrupts the whole system (Cristian, 1991) *e.g.*, in Kerlirzin & Vallet (1993), up to 8 neurons were destroyed in the experiments.

- **Weight Outliers:** For projection weights $W \in \mathbb{R}^{d_{\text{out}} \times d_{\text{in}}}$, $\tau$-weight outliers $O_{\text{weight}}^{\tau}$ are defined as:

$$O_{\text{weight}}^{\tau} = \{(i,j) \mid |W_{i,j}| > \tau \cdot \mu_{W_i}\},$$

where $\mu_{W_i} = \frac{1}{d_{\text{in}}} \sum_j |W_{i,j}|$ is the row-wise mean absolute value of $W$. We target in particular the weight tensor of the second projection layer in the MLP within each transformer block (An et al., 2025).

- **Attention Outliers:** For cumulative attention scores $A \in \mathbb{R}^{B \times H \times T \times T}$, $\tau$-attention outliers $O_{\text{attention}}^{\tau}$ are defined as:

$$O_{\text{attention}}^{\tau} = \{(b,h,j) \mid \hat{A}_{b,h,j} > \tau \cdot \mu_{A_{b,h}}\},$$

where $\hat{A}_{b,h,j} = \sum_{i=1}^{T} A_{b,h,i,j}$ is the cumulative attention contribution for token $j$, and $\mu_{A_{b,h}} = \frac{1}{T} \sum_j \hat{A}_{b,h,j}$ is the mean cumulative attention across keys. The random variable $\hat{A}_{b,h,j}$ is indexed on the keys and accounts for the total attention the j-th key receives from all the queries of the sequence. As such, attention outliers set contains the keys that receive overly large attention, far beyond the average. In a way, this formulation generalizes the first token predominance studied in Kaul et al. (2025).

## 3 RESULTS

In this section, we describe the negative results on the inefficiency of Dropout in fully suppressing the outliers. First, we study a miniature training of Tiny GPT-2 toy models of 6 blocks and 6 heads on shakespeare-char dataset (Caldas et al., 2018). We surgically modify the baseline training with different combinations of architecture and optimization related choices for two different Dropout rates. We include more details on our experimental setup, metrics and the Dropout layers' placement across the architectures we study in Section C and on our training policies in Section D. Herein, we provide a brief summary of our observations. The relative results are reported in Table 1.

We observe that, on average when using AdamW (Loshchilov & Hutter, 2019), Dropout training does reduce the outliers percentage as well as the excess kurtosis[2] and the Max-to-Median Ratio (MMR), most of the time for all the categories. In SGD and Muon (Jordan, 2024) trainings, we hardly observe any difference across the metrics tracked between the Dropout-ON and OFF setups (we overlook the loss in performance here). The choice of Positional Embedding (PE) (absolute (learnable) versus No Positional Embedding (NoPE) and Rotary Position Embedding (RoPE)) makes a difference as far as the impact of Dropout is concerned. Its effect is more perceptible for the absolute PE choice (baseline) and NoPE, but is negligeable when opting for RoPE. The impact of Weight Decay (WD) seems to be cumulative with that of Dropout. When disabling WD, the outliers' metrics decrease in the Dropout ON setting. When substituting LayerNorms (Section B.2) with RMSNorms, outliers noticeably decrease as well in Dropout training. Whereas, when completely disabling the Normalization layers and replacing them with Dynamic Tanh (Zhu et al., 2025), we hardly perceive the effect of Dropout training except on the activation outliers' metrics. Overall, we highlight that across all the combinations tested we never achieve complete suppression of these outliers while solely relying on Dropout training. Additional experiments and explanatory details are included in Section E.

We conduct additional experiments on a Tiny BERT (bidirectional architecture) model which we train for 5K iterations on Wikitext-103, sweeping over two Dropout rates again. In this experiment (Table 2), the model particularly underfits the dataset. The usage of Dropout does not help, but rather worsens the outliers' metrics. We believe this is due to the overregularization where the model creates outliers to compensate for the heavy reduction in its capacity.

Finally, we train a Tiny ViT model on Cifar-10 for 5K iterations. Overlooking the loss in accuracy, we observe again in this experiment (Table 3) that Dropout training decreases the outliers-relative metrics without achieving a complete suppression.

---

[2]With respect to a Gaussian distribution, for which the kurtosis is 3.

| Dropout rate | Optimizer | WD | Softmax | LN ( B.2) | PE | Attention | | | Activation | | | Weights | | | PPL |
|---|---|---|---|---|---|---|---|---|---|---|---|---|---|---|---|
| | | | | | | Kurtosis | Outliers(%) | MMR | Kurtosis | Outliers(%) | MMR | Kurtosis | Outliers(%) | MMR | |
| 0.0 | AdamW | 0.1 | Softmax | LN | Absolute(Learnable) | $17,843_{\pm7,755}$ | $2,997_{\pm1,077}$ | $253,187_{\pm161,416}$ | $7,763_{\pm4,311}$ | $1,150_{\pm0,479}$ | $24,983_{\pm1,606}$ | $1,167_{\pm0,205}$ | $0,197_{\pm0,012}$ | $7,213_{\pm0,045}$ | $4,580_{\pm0,076}$ |
| | SGD | 0.1 | Softmax | LN | Absolute(Learnable) | $4,340_{\pm0,000}$ | $0,780_{\pm0,000}$ | $8,860_{\pm0,000}$ | $-0,053_{\pm0,119}$ | $0,001_{\pm0,001}$ | $6,473_{\pm0,256}$ | $-1,207_{\pm0,006}$ | $0,006_{\pm0,001}$ | $5,270_{\pm0,020}$ | $21,637_{\pm0,006}$ |
| | Muon | 0.1 | Softmax | LN | Absolute(Learnable) | $10,890_{\pm0,411}$ | $0,933_{\pm0,071}$ | $11,247_{\pm0,762}$ | $0,600_{\pm0,044}$ | $0,113_{\pm0,012}$ | $10,523_{\pm0,468}$ | $0,280_{\pm0,000}$ | $0,033_{\pm0,006}$ | $5,977_{\pm0,015}$ | $5,593_{\pm0,059}$ |
| | AdamW | 0.0 | Softmax | LN | Absolute(Learnable) | $21,603_{\pm4,809}$ | $2,743_{\pm0,257}$ | $1,630 \cdot 10^{12}_{\pm1,787 \cdot 10^{12}}$ | $13,697_{\pm2,406}$ | $1,630_{\pm0,159}$ | $29,830_{\pm4,122}$ | $1,093_{\pm0,538}$ | $0,180_{\pm0,115}$ | $7,153_{\pm0,928}$ | $4,603_{\pm0,087}$ |
| | AdamW | 0.1 | Softmax | RMSNorm | Absolute(Learnable) | $18,673_{\pm3,668}$ | $2,703_{\pm0,568}$ | $1,490 \cdot 10^{12}_{\pm2,581 \cdot 10^{12}}$ | $10,747_{\pm1,605}$ | $1,273_{\pm0,159}$ | $27,310_{\pm5,257}$ | $1,263_{\pm0,211}$ | $0,200_{\pm0,040}$ | $7,193_{\pm0,326}$ | $4,587_{\pm0,055}$ |
| | AdamW | 0.1 | Softmax | DyT | Absolute(Learnable) | $12,267_{\pm6,610}$ | $1,177_{\pm0,350}$ | $3,710 \cdot 10^{10}_{\pm6,427 \cdot 10^{10}}$ | $-0,423_{\pm0,021}$ | $1,763_{\pm1,450}$ | $20,460_{\pm12,730}$ | $0,143_{\pm0,371}$ | $0,087_{\pm0,001}$ | $5,160_{\pm0,706}$ | $17,420_{\pm9,530}$ |
| | AdamW | 0.1 | Softmax | LN | RoPE | $5,853_{\pm0,530}$ | $0,697_{\pm0,021}$ | $16,323_{\pm3,803}$ | $6,817_{\pm1,845}$ | $0,407_{\pm0,015}$ | $26,150_{\pm2,012}$ | $0,330_{\pm0,080}$ | $0,043_{\pm0,015}$ | $5,933_{\pm0,122}$ | $5,103_{\pm0,114}$ |
| | AdamW | 0.1 | Softmax | LN | NoPE | $31,030_{\pm1,725}$ | $2,510_{\pm0,340}$ | $7,550 \cdot 10^{12}_{\pm6,564 \cdot 10^{12}}$ | $13,767_{\pm5,719}$ | $1,810_{\pm0,202}$ | $30,067_{\pm5,639}$ | $1,807_{\pm0,286}$ | $0,307_{\pm0,055}$ | $7,910_{\pm0,329}$ | $4,523_{\pm0,015}$ |
| 0.2 | AdamW | 0.1 | Softmax | LN | Absolute(Learnable) | $15,580_{\pm3,795}$ | $1,510_{\pm0,574}$ | $225,590_{\pm358,144}$ | $1,783_{\pm0,681}$ | $0,350_{\pm0,154}$ | $19,800_{\pm0,930}$ | $0,540_{\pm0,166}$ | $0,080_{\pm0,027}$ | $6,310_{\pm0,251}$ | $4,887_{\pm0,562}$ |
| | SGD | 0.1 | Softmax | LN | Absolute(Learnable) | $4,340_{\pm0,000}$ | $0,780_{\pm0,000}$ | $8,860_{\pm0,000}$ | $-0,022_{\pm0,117}$ | $0,001_{\pm0,001}$ | $6,463_{\pm0,290}$ | $-1,207_{\pm0,006}$ | $0,006_{\pm0,001}$ | $5,270_{\pm0,010}$ | $21,760_{\pm0,010}$ |
| | Muon | 0.1 | Softmax | LN | Absolute(Learnable) | $10,603_{\pm0,533}$ | $0,737_{\pm0,023}$ | $10,083_{\pm0,341}$ | $0,547_{\pm0,067}$ | $0,100_{\pm0,027}$ | $9,397_{\pm0,553}$ | $0,140_{\pm0,000}$ | $0,020_{\pm0,000}$ | $5,673_{\pm0,006}$ | $7,427_{\pm0,040}$ |
| | AdamW | 0.0 | Softmax | LN | Absolute(Learnable) | $15,617_{\pm2,998}$ | $1,423_{\pm0,450}$ | $217,497_{\pm280,282}$ | $2,133_{\pm1,140}$ | $0,463_{\pm0,232}$ | $20,970_{\pm2,757}$ | $0,537_{\pm0,170}$ | $0,080_{\pm0,030}$ | $6,313_{\pm0,260}$ | $4,863_{\pm0,527}$ |
| | AdamW | 0.1 | Softmax | RMSNorm | Absolute(Learnable) | $16,760_{\pm3,091}$ | $1,360_{\pm0,245}$ | $46,187_{\pm37,122}$ | $1,660_{\pm0,628}$ | $0,347_{\pm1,148}$ | $20,490_{\pm5,548}$ | $0,517_{\pm0,267}$ | $0,080_{\pm0,044}$ | $6,283_{\pm0,378}$ | $4,483_{\pm0,179}$ |
| | AdamW | 0.1 | Softmax | DyT | Absolute(Learnable) | $13,040_{\pm11,681}$ | $1,460_{\pm0,747}$ | $308,107_{\pm359,922}$ | $-0,777_{\pm0,317}$ | $0,886_{\pm0,809}$ | $13,543_{\pm8,347}$ | $-0,057_{\pm0,200}$ | $0,039_{\pm0,028}$ | $4,857_{\pm0,446}$ | $19,260_{\pm7,895}$ |
| | AdamW | 0.1 | Softmax | LN | RoPE | $6,147_{\pm0,574}$ | $0,670_{\pm0,046}$ | $23,730_{\pm3,450}$ | $4,490_{\pm0,988}$ | $0,523_{\pm0,065}$ | $22,483_{\pm1,369}$ | $0,323_{\pm0,035}$ | $0,047_{\pm0,006}$ | $5,917_{\pm0,006}$ | $4,277_{\pm0,038}$ |
| | AdamW | 0.1 | Softmax | LN | NoPE | $19,773_{\pm3,431}$ | $1,740_{\pm0,102}$ | $61\,511,590_{\pm97\,123,492}$ | $2,430_{\pm0,210}$ | $0,527_{\pm0,025}$ | $24,127_{\pm0,794}$ | $0,867_{\pm0,015}$ | $0,123_{\pm0,006}$ | $6,723_{\pm0,012}$ | $5,057_{\pm0,046}$ |

Table 1: Tiny GPT-2 Experiments post-training for 3K iterations averaged over transformer layers.

| Dropout rate | Attention | | | Activation | | | Weights | | | PPL |
|---|---|---|---|---|---|---|---|---|---|---|
| | Kurtosis | Outliers(%) | MMR | Kurtosis | Outliers(%) | MMR | Kurtosis | Outliers(%) | MMR | |
| 0.0 | $33,943_{\pm3,067}$ | $2,997_{\pm0,045}$ | $1,692 \cdot 10^{13}_{\pm7,345 \cdot 10^{11}}$ | $0,050_{\pm0,017}$ | $0,009_{\pm0,001}$ | $7,817_{\pm0,120}$ | $0,020_{\pm0,000}$ | $0,008_{\pm0,000}$ | $5,320_{\pm0,010}$ | $80,670_{\pm0,901}$ |
| 0.2 | $50,277_{\pm2,800}$ | $3,243_{\pm0,074}$ | $2,523 \cdot 10^{13}_{\pm1,336 \cdot 10^{12}}$ | $0,353_{\pm0,252}$ | $0,057_{\pm0,040}$ | $8,623_{\pm0,484}$ | $0,030_{\pm0,000}$ | $0,010_{\pm0,000}$ | $5,377_{\pm0,015}$ | $399,713_{\pm22,589}$ |

Table 2: Tiny BERT Experiments post-training for 5K iterations averaged over transformer layers.

| Dropout rate | Attention | | | Activation | | | Weights | | | Top1 Acc(%) |
|---|---|---|---|---|---|---|---|---|---|---|
| | Kurtosis | Outliers(%) | MMR | Kurtosis | Outliers(%) | MMR | Kurtosis | Outliers(%) | MMR | |
| 0.0 | $43,197_{\pm1,089}$ | $3,640_{\pm0,147}$ | $1,471 \cdot 10^{13}_{\pm3,346 \cdot 10^{12}}$ | $31,193_{\pm10,473}$ | $1,623_{\pm0,095}$ | $38,017_{\pm8,691}$ | $0,283_{\pm0,081}$ | $0,033_{\pm0,015}$ | $5,663_{\pm0,156}$ | $42,873_{\pm1,045}$ |
| 0.2 | $40,983_{\pm0,636}$ | $3,773_{\pm0,146}$ | $1,336 \cdot 10^{13}_{\pm2,446 \cdot 10^{12}}$ | $17,373_{\pm7,636}$ | $1,363_{\pm0,134}$ | $26,470_{\pm7,092}$ | $0,059_{\pm0,053}$ | $0,009_{\pm0,001}$ | $5,390_{\pm0,099}$ | $35,727_{\pm2,197}$ |

Table 3: Tiny ViT Experiments post-training for 5K iterations averaged over transformer layers.

## 4 CONCLUDING REMARKS

Our work suggests that several architecture and optimization related choices when training transformer-based models favour the formation of outliers and are antagonist with Dropout's expected implicit bias (Mianjy et al., 2018), that is known (in non-transformer based models) to have a rebalancing effect. In transformers, Dropout is not sufficient to compensate for the outliers formation, that is due to the optimizer, Normalization layers, Positional Emebddings, Softmax operator, etc. We refer to Section A for more details on outliers in transformer-based models.

To go one step beyond, one could draw a parallel with the sparse recovery and matrix completion literature (Chen et al., 2014): with the same logic that a uniform sampling strategy would miss most of the signal, one could expect that Dropout does not easily catch these extremely localized outliers. In that case, the outliers are not dropped frequently enough and their magnitude is not effectively dampened.

Our observations on transformer-based models across the setups we tested contrast with the insights given in the single hidden-layer linear neural networks in Mianjy et al. (2018). This raises important questions on the bias of Dropout in transformer-based models.

## 5 USE OF AI TOOLS

We use Large Language Models only to assist with coding and debugging. All the code was subsequently reviewed by the authors.

## 6 REPRODUCIBILITY STATEMENT

To ensure reproducibility, we include in Section C and Section D details on our experimental setups and the set of hyperparameters we use.

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

## A  RELATED WORKS

**Outliers in the quantization literature.**  Activation and weights outliers are mostly studied in the quantization literature. There, outliers are problematic as they widen the quantization range, resulting in potential performance loss (Tseng et al., 2024; Dettmers et al., 2022; Chee et al., 2024). These outliers have been defined either as "massive activations", that focus on per-element activation magnitudes (Sun et al., 2024; Kaul et al., 2025; An et al., 2025), or concurrently as "outliers features", restricted to feature dimension (Dettmers et al., 2022; He et al., 2024).

Recently, Vyas et al. (2025); Park et al. (2025); Vlassis et al. (2025); Nelson et al. (2023) considered the role of the optimizer's choice on the emergence of the activation outliers; they notably distinguished between diagonal preconditioners (Adam (Kingma & Ba, 2017), AdamW (Loshchilov & Hutter, 2019)) traditionally used for transformer-based models' training (Kunstner et al., 2024; Zhang et al., 2024), and non-diagonal preconditioners (Soap by Vyas et al. (2025), Muon by Park et al. (2025), Orthoadam by Kaul et al. (2025)). Other architectural choices' impact, namely Normalization layers and residual connections are analyzed in Kaul et al. (2025); He et al. (2024)

**Attention sinks.**  Attention outliers in An et al. (2025) or concurrently Attention sinks (Xiao et al., 2024; Yu et al., 2024b; Cancedda, 2024; Barbero et al., 2025) are also commonly observed in transformer-based models. Gu et al. (2025) summarize the impact of several architectural (softmax versus different kernels for attention, positional encoding) and optimization choices (weight decay) on sinks emergence. Similarly, the impact of softmax and its variants was considered in (Miller, 2023; Kaul et al., 2025; Ye et al., 2025; Saada et al., 2025; Zuhri et al., 2026; An et al., 2025; Qiu et al., 2025; Son et al., 2024). Ruscio et al. (2025) considers the choice of the positional encoding strategy, such as Absolute (Sinusoidal or Learned) versus relative (RoPE, NTK-aware RoPE) embeddings on the geometry of the attention's mechanism vector space.

From attention to activation outliers, the mechanism that links these categories is still unclear; some works argue for a correlation (Sun et al., 2024; Zhang et al., 2025; An et al., 2025), while others argue for a decorrelation (Kaul et al., 2025; Park et al., 2025) where one could persist while suppressing the other.

**Dropout.**  The standard Dropout formulation consists in randomly setting hidden activations in a given layer to zero (Srivastava et al., 2014; Baldi & Sadowski, 2013). Other variants include DropConnect (Wan et al., 2013), Dropattention (Zehui et al., 2019; Wang et al., 2024), Dropkey (Li et al., 2023) as customization of Dropout for transformer models, and stochastic-depth (Huang et al., 2016) in Vision Transformers. Oh et al. (2025) propose a dropout-based strategy to alleviate the model's reliance on outlier weights during a Low Rank Adaptation (LoRA) fine-tuning pipeline.

Although most of the recent pretrained models are trained without dropout, some works do use (implicitly) non-zero dropout rates to train the models they study the outlier issue on (Kovaleva et al., 2021). In particular in the OPT experiments of He et al. (2024), Dropout coupled with the architectural changes proposed yields better results.

## B  RECALLS ON TRANSFORMERS

### B.1  THE TRANSFORMER ARCHITECTURE

In this section, we review the standard transformer architecture components. Let $T$ denote the sequence length, $|V|$ the vocabulary size, $d$ the hidden dimension, $L$ the number of transformer blocks, $H$ the number of attention heads, $d_h = \frac{d}{H}$ and $d_{\text{up}}$ the inner dimension of the feed-forward network (FFN). We denote by $X$ the input token sequence.

We use the following notations: $\text{LN}(\cdot)$ for the normalization layer (Appendix. B.2), $\text{MHA}(\cdot)$ for the multi-head self-attention, and $\text{FFN}(\cdot)$ to denote a two-layer MLP with a nonlinearity, e.g., GELU. The attention mask is $M \in \mathbb{R}^{T \times T}$.

**Input embedding.**  The input is embedded and summed with positional embeddings:

$$H^0 \in \mathbb{R}^{T \times d} := XW_E + P,$$

where $W_E \in \mathbb{R}^{|V| \times d}$ is the learnable token embedding matrix and $P \in \mathbb{R}^{T \times d}$ is the Positional Embedding (PE).

We write here the default absolute positional embeddings formulation. P could be learnable as well or static (Sinusoidal PE of Vaswani et al. (2017)). The absolute embeddings are by default used in the original GPT2 version as well as in ViT and BERT models. A recently introduced alternative is the Rotary Positional Embedding (RoPE) (Su et al., 2024) which is an attention space positional encoding, we will recall later on. However, when using RoPE,

$$H^0 \in \mathbb{R}^{T \times d} := XW_E,$$

**Transformer blocks.**  For $l = 1, \ldots, L$ we write the block outputs as

$$H^l = [h_1^l, \ldots, h_T^l]^\top \in \mathbb{R}^{T \times d}.$$

Each block contains an MHA layer followed by an FFN. Next, depending on the placement chosen for the normalization layers, we could have two different block variants.

**Post-normalization layer** (the version proposed in the original transformer model by (Vaswani et al., 2017)) and in BERT models:

$$O^l = \text{LN}\big(\text{MHA}\big(H^{l-1}\big) + H^{l-1}\big),$$
$$H^l = \text{LN}\Big(\text{FFN}\big(O^l\big) + O^l\Big).$$

**Pre-normalization layer** (commonly used in autoregressive models like GPT and Llama):

$$O^l = \text{MHA}\big(\text{LN}(H^{l-1})\big) + H^{l-1},$$
$$H^l = \text{FFN}\big(\text{LN}(O^l)\big) + O^l.$$

**Multi-head self-attention (MHA).**  For each head $h = 1, \ldots, H$ the input $H^{l-1} \in \mathbb{R}^{T \times d}$ is linearly projected to queries, keys and values:

$$Q^{l,h} = H^{l-1}W_Q^{l,h}, \quad K^{l,h} = H^{l-1}W_K^{l,h}, \quad V^{l,h} = H^{l-1}W_V^{l,h},$$

with $W_Q^{l,h}, W_K^{l,h}, W_V^{l,h} \in \mathbb{R}^{d \times d_h}$.

If we use RoPE embeddings, an additional step consists then in rotating the key K and the query Q tensors such that :

$$K^{l,h} \mapsto R_\theta K^{l,h} \quad \text{and} \quad Q^{l,h} \mapsto R_\theta Q^{l,h}$$

Where $\theta$ is a hyperparameter that controls the frequency of rotation.

The attention scores are

$$A^{l,h} = \texttt{Softmax}\left(\frac{Q^{l,h}(K^{l,h})^\top}{\sqrt{d_h}} + M\right) \in \mathbb{R}^{T \times T},$$

where softmax is applied across each row (i.e., over key positions). The per-head outputs are $A^{l,h}V^{l,h} \in \mathbb{R}^{T \times d_h}$. Concatenating heads and projecting yields the output of the self attention block

$$\texttt{Concat}_{h=1}^{H}\left(A^{l,h}V^{l,h}\right)W_O^l,$$

with $W_O^l \in \mathbb{R}^{d \times d}$.

For causal (autoregressive) attention the mask $M$ is

$$M_{ij} = \begin{cases} -\infty, & i < j, \\ 0, & i \geq j. \end{cases}$$

**Feed-forward network (FFN).** The FFN is a two-layer MLP:

$$\text{FFN}(X) = \phi(XW_1 + b_1)W_2 + b_2,$$

with $W_1 \in \mathbb{R}^{d \times d_{\text{up}}}$, $W_2 \in \mathbb{R}^{d_{\text{up}} \times d}$, biases $b_1, b_2$, and nonlinearity $\phi$ (*e.g.*, GELU for GPT-2 models). The first layer projects in an upper dimension, while the second down-projects back the vector to the input dimension.

**Output.** The final block output $H^L$ is normalized and projected to vocabulary logits:

$$Y = \text{LN}(H^L)\,W_{\text{cls}}, \qquad W_{\text{cls}} \in \mathbb{R}^{d \times |V|},$$

Transformer-based models are usually trained using adaptive optimizers, mainly Adam or a variant thereof such as AdamW. These optimizers are preferred to plain SGD, as they have been shown to converge better for transformers. (Kunstner et al., 2024; Zhang et al., 2024). However, recently Muon (Jordan, 2024) usage, to update matrix parameters with AdamW as auxilary for the remaining parameters, proved to beat a new baseline achieving state-of-the-art results in LLM training (Liu et al., 2025b).

## B.2 NORMALIZATION LAYERS

In general, for a centring scalar $c \in \{0, 1\}$, a Normalization layer maps $\mathbf{x}$ to:

$$\texttt{Norm}(\mathbf{x}) = \frac{\mathbf{x} - c\,\mu(\mathbf{x})}{\sigma(\mathbf{x})} \odot \gamma + \beta,$$

where

$$\mu(\mathbf{x}) = \frac{1}{d}\sum_{i=1}^{d} x_i, \quad \sigma^2(\mathbf{x}) = \frac{1}{d}\sum_{i=1}^{d}\left(x_i - c\,\mu(\mathbf{x})\right)^2$$

Layer Norm (LN) corresponds to the case $c = 1$, with a trainable scale vector $\gamma$ and bias vector $\beta$, initialised to all ones and zeros respectively. While RMSNorm corresponds to setting $c = 0$ and $\beta = 0$ (He et al., 2024).

**Dynamic Tanh (DyT)** is a drop-in replacement for Normalization layers in transformers proposed recently in Zhu et al. (2025), and which writes as:

$$\texttt{DyT}(x) := \tanh\left(\alpha x\right) \odot \gamma + \beta$$

where $\alpha$ is a trainable parameter, initialized for attention layers with $\alpha_0 = 0.8$ and for MLP layers with $\alpha_0 = 0.2$ across Table 1 experiments.

## C EXPERIMENTAL PROTOCOL

We train three transformer-based models of six blocks and six heads, with different architectures (bidirectional and autoregressive) and on different modalities (image and text). In some cases we also test different optimizers (SGD, AdamW (Loshchilov & Hutter, 2019), Muon (Jordan, 2024)), and different training hyperparameters combinations (Dropout rate, weight decay). In what follows, we provide a summary of our experimental protocol.

**Models and datasets.** First, we study a causal architecture: *Tiny GPT-2 model*, built upon the repository of Karpathy, to pretrain our customized models on a next token generation task. GPT-2 is a decoder-only architecture that originally combines pre-layer norms, no biases, absolute learnable positional embeddings and softmax attention. We surgically modify the architecture in the experiments and discuss the interaction of each choice with Dropout training.

We also study a bidirectional architecture:

- *Tiny Bert model*, built upon the repository of zelindai, which we pretrain using Masked Language Modeling (MLM). BERT is an encoder-only architecture that combines post-layer norms, biases, absolute learnable positional embeddings and softmax attention.
- *Tiny ViT model* which we pretrain on an image classification task. ViT is also an encoder-only architecture combining pre-layer norms, with biases everywhere except in attention tensors( key, query and value) and softmax attention.

We use the Shakespeare-char dataset from the benchmark (Caldas et al., 2018) for the miniature Tiny GPT-2 pretraining, Wikitext-103 (Merity et al., 2016) for Tiny Bert and cifar-10 (Krizhevsky, 2009) for the experiments of ViT.

We report the detailed set of hyperparameters used for training in Appendix. D and we include details on the tokenizers used.

**Metrics.** Prior to conducting our experimental evaluation, We extend the definitions of Section 2.2 and monitor the evolution of other metrics on $H_{i,j,k}$, $W_{i,j}$ and $\hat{A}_{b,h,j}$ respectively. Hence, besides looking at outliers percentages for $\tau = 5$ in our experiments, we monitor the *excess kurtosis* of the distributions of the elements of the tensors of interest, to quantify the degree of outlier concentration with respect to a Gaussian[3]. Moreover, we monitor the evolution of the Maximum-to-Median Ratio (MMR) across training for all the tensors of interest.

Finally, we also report for BERT and GPT-2 experiments the pre-training Perplexity(exponential of pre-training test-set loss) and the Top1 accuracy for ViT experiments.

These metrics are measured in Eval mode of the model over a subsample of the test-set. Hence, the Dropout layers are deactivated and no node is masked for the evaluation.

**Dropout layers placement.** Default placements of Dropout layers exist in the architectures we use, where enabling Dropout is controlled through the Dropout rate that defines its strength. The standard positions are respectively: Dropout on the attention map, Dropout at the output of the attention block, Dropout at the output of the FFN block and a dropout after the Embedding layer. With some particularity of ViT models that could place an additional dropout layer after the activation of the first linear projection inside the FFN block.

For simplicity, we adopt shared Dropout rate values across the models in all our experiments.

## D HYPERPARAMETERS

Additionnally, for ViT models, we use the following augmentation strategies for training on Cifar-10: RandomHorizontalFlip, RandomCrop .

In all the experiments, we use a cosine annealing schedule for the optimizer.

We provide results averaged across three different seeds (1337, 42 and 573) in Table 1, Table 8, Table 2 and Table 3.

---

[3]

$$\text{Excess} - \text{Kurt}[A] = \mathbb{E}\left[\left(\frac{A - \mu}{\sigma}\right)^4\right] - 3,$$

where $\mu$ and $\sigma$ denote the mean and standard deviation of the elements in tensor $A$, respectively. And 3 is the kurtosis value associated to a Gaussian distribution.

Table 4: Hyperparameters used for training Tiny GPT-2 model on `Shakespeare-char`

| Hyperparameter | Shakespeare-char |
|---|---|
| Eval iterations | 200 |
| Gradient accumulation steps | 1 |
| Batch size | 16 |
| Block size | 256 |
| Number of transformer layers | 6 |
| Number of attention heads | 6 |
| Embedding dimension | 384 |
| MLP expansion factor | 4 |
| Learning rate | $10^{-3}$ |
| Minimum learning rate | $10^{-4}$ |
| $\beta_2$ | 0.99 |
| Momentum | 0.9 |
| Warmup iterations | 100 |
| LR decay iterations | 3000 |
| Max iterations | 3000 |
| Weight Decay | 0.1 |
| $\tau$ | 5 |
| $\epsilon$ | $10^{-8}$ |
| ROPE ($\theta$) | 10000 |
| Gradient clip | 1 |

Table 5: Hyperparameters used for training Tiny BERT Model on `Wikitext-103`

| Hyperparameter | Wikitext-103 |
|---|---|
| Eval iterations | 200 |
| Gradient accumulation steps | 4 |
| Batch size | 16 |
| Block size | 256 |
| Number of transformer layers | 6 |
| Number of attention heads | 6 |
| Embedding dimension | 384 |
| MLP expansion factor | 4 |
| Masking probability | 0.15 |
| Learning rate | $5 \cdot 10^{-4}$ |
| Minimum learning rate | $5 \cdot 10^{-5}$ |
| $\beta_2$ | 0.99 |
| Momentum | 0.9 |
| Warmup iterations | 100 |
| LR decay iterations | 5000 |
| Max iterations | 5000 |
| Weight Decay | 0.1 |
| $\tau$ | 5 |
| $\epsilon$ | $10^{-8}$ |
| Gradient clip | 1 |

Table 6: Hyperparameters used for Tiny ViT model training on `Cifar-10`

| Hyperparameter | Cifar-10 |
|---|---|
| Eval iterations | 500 |
| Gradient accumulation steps | 1 |
| Batch size | 64 |
| image size | 32 |
| patch size | 4 |
| Number of transformer layers | 6 |
| Number of attention heads | 6 |
| Embedding dimension | 384 |
| MLP expansion factor | 4 |
| Learning rate | $10^{-3}$ |
| Minimum learning rate | 0 |
| $\beta_2$ | 0.99 |
| Momentum | 0.9 |
| Warmup iterations | 200 |
| LR decay iterations | 5000 |
| Max iterations | 5000 |
| Weight Decay | 0.01 |
| $\tau$ | 5 |
| $\epsilon$ | $10^{-8}$ |
| Gradient clip | 1 |

For GPT-2, we use a pretrained BPE tokenizer for the inputs from tiktoken library (Radford et al., 2019). For BERT, we use a pretrained BERT-base-cased tokenizer provided by the Hugging Face Transformers library (Krizhevsky, 2009).

**Details on Models initialization.** In the standard setup, we initialize GPT-2 models' weight parameters from a $\mathcal{N}(0, 0.02)$ with an additional rescaling on the last linear projections at the end of

the attention and the FFN blocks, thus initializing their weight matrix from a $\mathcal{N}(0, \frac{0.02}{\sqrt{2 \times L}})$. All bias vectors if relevant are initialized to zero.

BERT and ViT models are simply initialized from $\mathcal{N}(0, 0.02)$ and bias vectors set to zero.

## E   EXPERIMENTS

In this section, we provide further analysis of the results reported in Section 3 as well as some complementary experiments.

**Motivation.**    First, we validate our initial robustness motivation from Section 1. We evaluate 3K-iteration-trained Tiny GPT-2 best-performance checkpoints corresponding to the vanilla setup at zero Dropout in Table 1 over 200 randomly sampled sequences from the validation dataset, where we zero out all the different outliers categories as defined in Section 2.2 at once. For better comparison, we further compute the following baseline, where we **randomly** zero-out the same fraction of elements as the outliers' percentages.

Additionnally, we evaluate the Dropout-trained checkpoints' sensitivity to both ablation strategies.

All the results are reported in Table 7.

| Model | Ablation strategy | Perplexity (before) | Perplexity (after) |
|---|---|---|---|
| Dropout OFF | Random | $4{,}690_{\pm 0{,}078}$ | $7{,}313_{\pm 2{,}561}$ |
|  | Outliers | $4{,}690_{\pm 0{,}078}$ | $47{,}483_{\pm 36{,}043}$ |
| Dropout ON (0.2) | Random | $4{,}963_{\pm 0{,}572}$ | $4{,}970_{\pm 0{,}566}$ |
|  | Outliers | $4{,}963_{\pm 0{,}572}$ | $5{,}383_{\pm 0{,}578}$ |

Table 7: Sensitivity of models trained with (Dropout ON) and without (Dropout OFF) Dropout to the ablation of nodes randomly versus **outliers** in inference. The Dropout ON models used 0.2 Dropout rate.

We observe that the performance drops and the perplexity increases noticeably when zeroing out **the outliers** in the Dropout OFF trained models, despite the fraction of these outliers with respect to the total parameters of the model being very small.

We clearly observe that, in the Dropout OFF models, the ablation of the outliers results in a more drastic perplexity increase in comparison to the random selection and zeroing out of the same fraction of nodes across the model, thus highlighting the vulnerability of the model in the presence of these outliers.

Conversely, the Dropout ON trained models seem to be more resistant to the outliers' ablation. Hence, although ultimately the outliers were not fully suppressed as it was displayed in Table 1, we find that dropout training alleviates the reliance of the model on the outliers' presence. Hence, zeroing out these outliers results in much smaller decrease in performance in comparison to the model trained without Dropout.

In the Dropout ON models, we observe no difference when randomly ablating the same fraction of the nodes as the outliers' percentages, which is expected given the inherent functioning of Dropout's mechanism. In comparison, the effect of ablating the outliers in the Dropout ON models, while moderate, is still slightly above this random baseline.

**Extended results.**    In this section, we expand the analysis of the results reported in Tables 1 to 3 and 8. The architectural and optimization related combinations we test are mainly driven by the observation that certain training and design choices (including the optimizer as well as layernorms, weight decay and positional embedding) affect the different outliers' emergence and evolution during training (Gu et al., 2025; Ruscio et al., 2025; He et al., 2024).

In Table 1, we first observe that activations outliers' metrics are the most consistently reduced across all the design and training choices we test. Attention statistics seem also to improve but less uniformly

(slight increase of the kurtosis and attention outliers' percentage when using DyT for instance). However, the biggest improvement is in terms of MMR for attention outliers, with more moderate values and significantly less variability in the Dropout ON case in comparison to the Dropout OFF one. Weights are already less extreme than activations in the Dropout OFF setting, but Dropout ON still reduces weight outliers' metrics.

Overall, we notice that the strongest smoothing effect appears where the Dropout OFF setting was already somewhat unstable, especially for AdamW with normalization/positional choices that seem to permit spiky dynamics. Dropout then seems to help reduce marginally these spikes rather than fighting the whole optimization process. Whereas, Dropout seems to be ineffective in the RoPE, Muon and SGD settings. Under SGD, due to the rescaling factor applied with Dropout, tensors are invariant in expectation, which would explain the quasi-invariance observed on the metrics between the dropout ON and OFF setups. Muon and RoPE already non-trivially alter the geometry of the model and Dropout seems only to add a limited amount of extra regularization in these cases.

In most of the realistic training scenarios, the Dropout ON setting is accompanied by a loss in performance (observe Bert and ViT experiments), on which we do not focus in our work. This is in fact expected in the setups where we perform a single pass on the data, in which case there is no risk of overfitting (Liu et al., 2025a). Dropout's usage translates in a reduction of the capacity of the model and thus in a performance decrease.

**Dropout-aware initialization experiment.** In this experiment, we reproduce the baseline model changing the way we initialize the model by the Dropout-aware initialization proposed in Kedia et al. (2024). We do not apply the reweighting of the residual branches (we maintain our rescaling factor).

Embedding layers are initialized with $\mathcal{N}(0, \sigma^2_{\text{embd}})$ where $\sigma^2_{\text{embd}} = \frac{1-p}{2}$. All linear layers except for the query and key weight tensors that are initialized with $\mathcal{N}(0, \sigma^2_{ln})$ where $\sigma^2_{ln} = \frac{1}{n_{\text{emb}}}\sqrt{\frac{1-p}{2}}$. A second exception is also in the last projection layers of the attention/FFN blocks respectively where they get an additional rescaling $\sigma^2_{ln} = \frac{1}{n_{\text{emb}}}\sqrt{\frac{1-p}{4L}}$.

Query and key tensors are initialized with $\mathcal{N}(0, \sigma^2_{qk})$ where $\sigma^2_{qk} = \frac{1}{n_{\text{emb}}}$.

The relative results are reported in Table 8.

| Dropout rate | initialization | Attention | | | Activation | | | Weights | | | PPL |
|---|---|---|---|---|---|---|---|---|---|---|---|
| | | Kurtosis | Outliers(%) | MMR | Kurtosis | Outliers(%) | MMR | Kurtosis | Outliers(%) | MMR | |
| 0.0 | Vanilla | $17{,}843_{\pm7{,}755}$ | $2{,}997_{\pm1{,}077}$ | $253{,}187_{\pm161{,}416}$ | $7{,}763_{\pm4{,}311}$ | $1{,}150_{\pm0{,}479}$ | $24{,}983_{\pm1{,}606}$ | $1{,}167_{\pm0{,}205}$ | $0{,}197_{\pm0{,}012}$ | $7{,}213_{\pm0{,}045}$ | $4{,}580_{\pm0{,}076}$ |
| | Dropout-aware | $24{,}957_{\pm6{,}247}$ | $2{,}510_{\pm0{,}174}$ | $1{,}050 \cdot 10^8_{\pm1{,}791 \cdot 10^8}$ | $1{,}877_{\pm0{,}219}$ | $0{,}470_{\pm0{,}027}$ | $18{,}333_{\pm1{,}074}$ | $0{,}267_{\pm0{,}006}$ | $0{,}030_{\pm0{,}000}$ | $5{,}897_{\pm0{,}006}$ | $4{,}613_{\pm0{,}057}$ |
| 0.2 | Vanilla | $15{,}580_{\pm3{,}795}$ | $1{,}510_{\pm0{,}574}$ | $225{,}590_{\pm358{,}144}$ | $1{,}783_{\pm0{,}681}$ | $0{,}350_{\pm0{,}154}$ | $19{,}800_{\pm0{,}930}$ | $0{,}540_{\pm0{,}166}$ | $0{,}080_{\pm0{,}027}$ | $6{,}310_{\pm0{,}251}$ | $4{,}887_{\pm0{,}562}$ |
| | Dropout-aware | $25{,}160_{\pm1{,}452}$ | $1{,}270_{\pm0{,}046}$ | $24{,}927_{\pm11{,}827}$ | $3{,}437_{\pm0{,}543}$ | $0{,}523_{\pm0{,}032}$ | $20{,}033_{\pm1{,}845}$ | $0{,}223_{\pm0{,}012}$ | $0{,}030_{\pm0{,}000}$ | $5{,}820_{\pm0{,}036}$ | $4{,}690_{\pm0{,}040}$ |

Table 8: Tiny GPT-2 Experiments post-training for 3K iterations averaged over transformer layers.

When considering the second initialization strategy, we hardly observe any improvement on average between the dropout ON and OFF settings, except on MMR and attention outliers percentage metrics. However, when comparing the dropout trained model standardly initialized, we could note a slight decrease in the weight outliers metrics as well as some of the attention outliers' metrics (MMR and percentage except for the kurtosis) whereas activation outliers' metrics slightly increase. The dropout-aware version is not uniformly smoother; it mainly seems to shift the kind of structure the model learns, rather improving the balance of attention more than the overall smoothness.

**Targeted Dropout experiment.** In this experiment, we provide a comparison between Vanilla dropping strategy detailed in Section C and more targeted Dropout strategies: DropKey (Li et al., 2023) and DropConnect (Wan et al., 2013). The choice of these strategies in particular is derived by the definitions we work with for the outliers (Section 2.2). The results are reported in Table 9.

We observe a slight reduction the weights outliers metrics, but activation and attention outliers' metrics do increase significantly. Attention becomes extremely max-dominated, much more than in vanilla setup and highly inconsistent across runs (higher standard deviation). Overall, this strategy

| Dropout rate | Strategy | Attention | | | Activation | | | Weights | | | PPL |
|---|---|---|---|---|---|---|---|---|---|---|---|
| | | Kurtosis | Outliers(%) | MMR | Kurtosis | Outliers(%) | MMR | Kurtosis | Outliers(%) | MMR | |
| 0.0 | Vanilla | $17{,}843_{\pm 7{,}755}$ | $2{,}997_{\pm 1{,}077}$ | $253{,}187_{\pm 161{,}416}$ | $7{,}763_{\pm 4{,}311}$ | $1{,}150_{\pm 0{,}479}$ | $24{,}983_{\pm 1{,}606}$ | $1{,}167_{\pm 0{,}205}$ | $0{,}197_{\pm 0{,}012}$ | $7{,}213_{\pm 0{,}045}$ | $4{,}580_{\pm 0{,}076}$ |
| 0.2 | Vanilla | $15{,}580_{\pm 3{,}795}$ | $1{,}510_{\pm 0{,}574}$ | $225{,}590_{\pm 358{,}144}$ | $1{,}783_{\pm 0{,}681}$ | $0{,}350_{\pm 0{,}154}$ | $19{,}800_{\pm 0{,}930}$ | $0{,}540_{\pm 0{,}166}$ | $0{,}080_{\pm 0{,}027}$ | $6{,}310_{\pm 0{,}251}$ | $4{,}887_{\pm 0{,}562}$ |
| | DropConnect+DropKey+ DropEmb | $29{,}583_{\pm 14{,}331}$ | $3{,}577_{\pm 1{,}714}$ | $3{,}066 \cdot 10^{12}_{\pm 2{,}621 \cdot 10^{12}}$ | $9{,}187_{\pm 13{,}389}$ | $0{,}840_{\pm 0{,}490}$ | $23{,}770_{\pm 9{,}879}$ | $0{,}380_{\pm 0{,}076}$ | $0{,}050_{\pm 0{,}010}$ | $6{,}073_{\pm 0{,}100}$ | $6{,}517_{\pm 1{,}418}$ |

Table 9: Tiny GPT-2 Experiments post-training for 3K iterations averaged over transformer layers.

seems to introduce hard structural perturbations in the model, thus rather worsening the outliers' metrics.

