# OpenReview forum: "Dropout and the Outliers: Could Transformers Overcome Their Single Points of Failure?"
_ICLR.cc/2026/Workshop/Sci4DL — Sci4DL 2026_

### Official Review · Reviewer_P7r8 · 2026-02-27

**Fit:** 3
**Significance:** 2
**Confidence:** 2

**Summary:**

This paper studies whether dropout can mitigate the emergence of outliers in transformer training. The authors consider three outlier types (activation, attention, and weight outliers) defined via a threshold rule (with $\tau=5$) and track additional heavy-tail metrics such as excess kurtosis and max-to-median ratio. They run controlled experiments on small transformer models across modalities and architectures (Tiny GPT-2, Tiny BERT, Tiny ViT), while varying training and design choices including optimizer (AdamW/SGD/Muon), weight decay, positional embeddings (absolute/NoPE/RoPE), and normalization variants (LayerNorm/RMSNorm/DyT). The main result is that dropout often reduces outlier metrics only marginally (mostly under AdamW) and never fully suppresses outliers; in some settings (notably the Tiny BERT setup) dropout can even worsen outlier metrics, suggesting strong interactions between dropout and other architectural/optimization choices.

**Strengths:**

The paper targets a timely and practically relevant phenomenon: outliers in transformers that matter for robustness and deployment constraints (for example quantization and hardware sensitivity). The study is systematic in the sense that it sweeps multiple design axes (optimizer, positional embeddings, normalization, weight decay) rather than attributing effects to dropout in a single fixed pipeline. The metrics are clearly defined and cover complementary views of “outlierness” (threshold-based counts plus heavy-tail statistics). The experimental protocol and hyperparameters are described in sufficient detail to support reproducibility, and the negative or mixed results are reported transparently.

**Suggestions:**

A key limitation is scale: all experiments use tiny models and short training horizons, and many results may not transfer to modern large-scale pretraining. It would strengthen the paper to include at least one larger model (or a longer training regime) and to report variability across multiple random seeds, since outlier emergence can be unstable. In addition, the paper sometimes “overlooks the loss in performance”; it would be helpful to quantify the trade-off between reducing outlier metrics and maintaining task performance, and to report both training-time and evaluation-time statistics consistently. Finally, the paper frames outliers as “single points of failure”, but does not directly test failure modes; adding stress tests (for example ablation of outlier features, bit-flip sensitivity, or quantization robustness) would connect the observed metrics to the claimed robustness motivation. Mechanistically, a deeper analysis of why dropout becomes ineffective under certain choices (for example RoPE, Muon/SGD, or DyT) would improve the explanatory value; even a simple analysis of the probability that a localized outlier is dropped, or comparisons to targeted variants (dropkey/dropattention) would be informative.

---

### Official Review · Reviewer_Hz5m · 2026-02-27

**Fit:** 2
**Significance:** 1
**Confidence:** 2

**Summary:**

The authors study the effect of Dropout on outliers in small transformers.  In practice, a model is created with either no or 20% dropout for 3 different tasks (Tiny GPT-2 on shakespeare-char; Tiny Bert on Wikitext-103; Tiny ViT model on Cifar-10) and the effect on outliers (weight, activation and attention outliers) measured.

Conflicting results are observed dependant on settings, but in general Dropout does not seem to suppress outliers in transformers to the extent they do in other models.

**Strengths:**

This is an interesting and relevant question, and the general approach and metrics used are sound.

**Suggestions:**

The current execution is still very limited:
- The training process is not yet clearly described: Section D describes fixed parameters only, while Section C mentions sweeping over ‘Weight Decay and Dropout’ even though only 2 values of each seem to be tested. This is quite a limited setting.
- Do you have any evidence that the comparative results are insensitive to learning rate? It’s not clear to me that either the dropout=0 or dropout=0.2 models are well optimised (properly hyperparameter tuned). Without having tuned models, the comparison may be misleading.
- From Section D it seems the results in the tables are for a single seed. Given the small size of the models, it should be computationally feasible to rerun the experiments over more seeds, and to report on the variability across runs (e.g. standard error).
- Results are currently displayed without much analysis. Better substantiated results, better analysed, may produce more clearly observable patterns.

Minor:
- Introduce abbreviations like RoPE, NoPE even if standard.
- “ours setups”, “outliers features”

---

### Meta-Review · Area_Chair_Si5s · 2026-03-02

**Recommendation:** Accept

**Metareview:**

One reviewer points out a number of limitations that require revision of the paper. However, since the paper is a good fit for the workshp, I recommend accept. Authors should leverage the feedback to improve the paper before presentation.

---

### Decision · Program_Chairs · 2026-03-02

Accept